# Using full-text content to characterize and identify best seller books: A study of early 20th-century literature

**Giovana D. da Silva**[1], **Filipi N. Silva**[2]*, **Henrique F. de Arruda**[3], **Bárbara C. e Souza**[1], **Luciano da F. Costa**[4], **Diego R. Amancio**[1]

**1** Institute of Mathematics and Computer Science – USP, São Carlos, SP, Brazil, **2** The Observatory on Social Media (OSoMe), Indiana University, Bloomington, Indiana, United States of America, **3** CENTAI Institute, Turin, Italy, **4** São Carlos Institute of Physics – USP, São Carlos, SP, Brazil

* filsilva@iu.edu

## Abstract

Artistic pieces can be studied from several perspectives, one example being their reception among readers over time. In the present work, we approach this interesting topic from the standpoint of literary works, particularly assessing the task of predicting whether a book will become a best seller. Unlike previous approaches, we focused on the full content of books and considered visualization and classification tasks. We employed visualization for the preliminary exploration of the data structure and properties, involving SemAxis and linear discriminant analyses. To obtain quantitative and more objective results, we employed various classifiers. Such approaches were used along with a dataset containing (i) books published from 1895 to 1923 and consecrated as best sellers by the *Publishers Weekly Bestseller Lists* and (ii) literary works published in the same period but not being mentioned in that list. Our comparison of methods revealed that the best-achieved result—combining a bag-of-words representation with a logistic regression classifier—led to an average accuracy of 0.75 both for the leave-one-out and 10-fold cross-validations. Such an outcome enhances the difficulty in predicting the success of books with high accuracy, even using the full content of the texts. Nevertheless, our findings provide insights into the factors leading to the relative success of a literary work.

## 1 Introduction

Understanding the factors and reasons determining the effectiveness and acceptance of given pieces of artistic or scientific work represents a continuing challenge in artificial intelligence (e.g., [1–5]). As it is often the case with complex systems, not only a large number of possible factors is potentially involved, but their individual and combined effects also tend to be highly non-linear. In this manner, small effects can lead to considerable impacts, being also likely to vary along time and space in modes that are hard to predict.

Among the several aspects that are more likely to influence the visibility and accomplishment of an artistic piece, we have its intrinsic *quality*, *innovation*, and *affinity* with the main trends, interests, and expectations predominating in a given period and place. All these three

**Data Availability Statement:** The dataset of books were obtained from the Gutenberg dataset and are distributed in public domain. The specific subset

employed in this study can be found in http://dx.doi.org/10.5281/zenodo.7622473.

**Funding:** G. D. da S. acknowledges São Paulo Research Foundation (FAPESP) from Brazil for sponsorship (grant no. 2021/01744-0). B. C. e S. acknowledges Coordination for the Improvement of Higher Education (CAPES) from Brazil for sponsorship Finance Code 001. L. da F. C. thanks Brazilian National Council for Scientific and Technological Development (CNPq) (grant no. 307085/2018-0) and São Paulo Research Foundation (FAPESP) (grant grant 15/22308-2). D. R. A. acknowledges financial support from Brazilian National Council for Scientific and Technological Development (CNPq) (grant no. 311074/2021-9) and São Paulo Research Foundation (FAPESP) (grant no. 2020/06271-0). The funders had no role in study design, data collection and analysis, decision to publish, or preparation of the manuscript.

**Competing interests:** The authors have declared that no competing interests exist.

main aspects are not only challenging to *define* but even more so to *predict*, which has motivated growing interest from the scientific community (e.g., [6–11]).

A better understanding of the motivations why an artistic piece becomes successful constitutes a particularly interesting objective for a handful of reasons: (i) this type of study can motivate the development of new concepts and methods capable of quantifying the three main aspects identified above, namely quality, innovation, and affinity of an artistic piece; (ii) that kind of research has great potential for revealing important aspects of the mechanisms underlying human preferences for specific subjects and styles along time and space; (iii) such developments can lead to strategies for predicting the acceptance of certain types of works, which may provide subsidies and motivation for developing new and more effective artistic pieces.

The present work aims at studying whether it is feasible to characterize and identify stories and narratives listed as best sellers by combining full-text content information and machine learning models. In this regard, the textual content of a set of books was modeled, and a series of experiments assessed the possibility of automatically differentiating a best seller from an ordinary book. In particular, we employed a dataset encompassing the full-text content of literary works collected from the Project Gutenberg platform. The dataset was split into two categories: *success* (books that appear at least once in the *Publishers Weekly Bestseller Lists*) and *others*. After applying a preprocessing step (removal of stopwords, lemmatization, and tokenization), the content of each book was embodied in terms of a word embedding representation by using the bag-of-words [12] and doc2vec [13] approaches. Finally, we employed different strategies to assess the prediction of the success of books in terms of their embedding representations, including: (i) visualization approaches, namely the linear discriminant analysis (LDA) [14] and SemAxis [15] techniques; and (ii) classification approaches, encompassing different models and cross-validation strategies.

In contrast to previous studies, here we rely on one of the prime published sources of best sellers book lists, namely the *Publishers Weekly Bestsellers Lists*, which comprises the best selling books every year since 1895. Although its criteria to define a book as an absolute success is not entirely specified, it is established that every considered paperbound book sold at least 2,000,000 copies, and every selected hardbound book sold 750,000 copies or more. It is also settled that Publishers Weekly only regards books distributed through the trade—that is, bookstores and libraries –, not including those sold by mail or book clubs [16]. Besides that, our work compounds the list of few studies which analyzed the success factor by analyzing the full-text content of the texts, posthumously modeling it through embeddings, and analyzing it both qualitatively (applying visualization and seeking for words that lead to discrimination) and quantitatively (involving supervised classifiers).

Ultimately, the results obtained from the considered approaches using only a book's full-text content were insufficient to predict the success of a literary work with high accuracy. The best classification accuracy achieved the value of 0.75 by combining a bag-of-words representation with a logistic regression model, which is a fair-to-middling outcome. Nonetheless, our experiments evince that the subject (literary genre provided by Gutenberg) of a book, alone, does not seem to be enough to determine if a title will become a best seller, but rather point to the importance of content, since there are words there are more typically found in this category of books.

This work is organized as follows. Section 2 presents and discusses the related works. In Section 3, we present the research questions. Section 4 describes the used datasets. Section 5 describes the methodology adopted to analyze the books, including text preprocessing, representation, visualization, and classification. The results and discussions are reported in Section 6. Finally, in Section 7, we present the conclusions and future works.

## 2 Related works

The study conducted in [17] analyzed the success of books using as reference the *The New York Times Best Sellers*, which includes a list of best selling books in the United States. The authors considered the books appearing on the list between August 2008 and March 2016. As additional information, the sales patterns of books were also considered by using data from *NPD BookScan* [17]. Several interesting results were reported. Fiction books were found to be more likely to become best sellers, while nonfiction books tended to be sold with lower intensity. The authors also proposed a model that can accurately measure long-term impact since it can predict the number of copies sold by best sellers short after their release. The proposed description was found to be consistent with a previous model devised to describe the attention received by scientific papers [1]. The authors argue, therefore, that the underlying processes of attention are similar—despite the differences in time scale.

A model to predict book sales was proposed in [6]. The authors used as a dataset the *NPD Bookscan*, focusing on a list of the 10 thousand top-selling books in a given period. A machine learning approach was proposed using different book features. Authors' visibility was taken into account by measuring the public interest in authors via Wikipedia page views. Previous sales were also considered as a feature to measure the previous success of authors. Book features included genre (e.g., horror and science fiction) and topic information (as provided by readers). In addition, publishers' information was used. All features were combined in the so-called Learning to Place (L2P) machine learning algorithm [18], which aims at classifying a new instance (i.e., predicting book sales) within a sequence of previously published books. This study found that in fiction and nonfiction books, the publisher quality tends to play an important role in the prediction. The visibility of authors was also found to be an important feature, as more visible authors potentially are more likely to sell more copies. Finally, the other factors related to the text content itself (e.g., genre and topic information) were found to play relatively a minor role in the prediction model.

Differently from previous works that did not take into account the textual content [6, 17], the relevance of writing style was analyzed in [19]. The authors analyzed full books from different genres (e.g. adventure, mystery, fiction). The dataset was collected from the *Project Gutenberg* repository. Several linguist marks of writing style were used to characterize the texts. Examples include lexical features, distribution of grammar rules, and sentiment analysis. The authors used SVM as classifier [20], and download counts were used as a surrogate for the visibility of books. Additional information such as award recipients and the number of copies sold was also used to quantify success. The authors concluded that the used stylistic metrics are effective to quantify the success of novels.

Because only a few works have analyzed the content of books to predict if they will become best sellers, in the current study we focus our analysis on full-textual features to discriminate between best sellers and ordinary literary works.

## 3 Research questions

This study aims to test whether the full-text content of the book alone can indicate if it will become a best seller. While there are several ways to represent a text, we focused on the most common approaches devoted to representing *long texts*. For this reason, here we also investigate which text representation better grasps the information about a book becoming a success. Finally, to recognize patterns in the textual data, we also examined which classifier is the most appropriate for discriminating between successes and ordinary books.

Briefly, the main research questions here are:

1. Is it possible to predict the inclusion of books into best sellers lists by analyzing only their full-text content?

2. Can one use bag-of-words and neural network embeddings to detect informative attributes for identifying best sellers?

3. Can the abovementioned embeddings be influenced by the subject headings available in the dataset (such as genre or literary class)?

4. How different is the performance of supervised classifiers in discriminating between the two categories of books analyzed?

## 4 Dataset

As the main objective of this work is to understand whether it is possible to identify and characterize styles and stories classified as best sellers, our dataset was composed of two categories: *success* and *others*. In the first, we included books considered best sellers; in the latter, literary works not listed as such (at least not in the analyzed period and the consulted list). All considered instances were written in English.

To define the candidate books for the *success* category, we resorted to well-known annual lists: *The New York Times Best Sellers*, first published in 1931, and *Publishers Weekly Bestseller Lists*, first published in 1895. Concerning the first one, from 1931 to the present day, only 18 titles were available on the *Project Gutenberg* platform (a digital library whose collection is composed of full texts of books in the public domain). For the second, we mapped 110 available titles—published from 1895 to 1923—which became part of our dataset.

To select the titles of the *other* category, we considered the collection of books (a) published in the same period as the selected successful ones and (b) not included in the best sellers lists of *Publishers Weekly*. In this sense, if the *success* class had ten titles published in 1923, the *other* would have the same number of titles published in the same year—the titles randomly selected from the *Gutenberg* repository. At the end of the process, this category contained 109 titles (one less than the other category, as it was infeasible to collect the same amount of titles for all the years considered).

In [16] can be found the best selling lists used in this study. It is important to emphasize that the criteria for composition on the list are not entirely clear. Every hardbound book in it has sold at least 750,000 copies, and every paperbound book has sold at least 2 million copies. There is no clarification as to why such numbers were chosen as the minimum quantity to define a best selling title. Besides that, only sales of books distributed in trade (bookstores and libraries) are accounted for, a criterion that excludes those sold by mail order or reading clubs. It is also not specified why only these specific sales were considered—a reasonable explanation being that since these are somewhat old books, it was not so easy to keep track of all kinds of markets.

Additionally, it is worth mentioning that some factors were imperative in the limited number of books of the dataset (namely, 219 instances). First, we adhere to titles in the public domain only. Although there are discussions about the fair use of such content in scientific works, there is no consensus on the validity of using copyrighted pieces. Second, we considered only one book from each author to avoid identification of authorship by machine learning algorithms to be applied later. Third, because one of the design decisions was to work with a balanced database, the number of bestsellers becomes a limiting factor for the number of non-bestselling books. Lastly, we collected the same number of successes and non-successes per year of publication (which even led to one less non-successful book due to the unavailability of another title in one of the years considered). We emphasize, nonetheless, that such a

temporal factor is essential because there will always be a possibility that titles from different periods may be very distinct in terms of content and writing style. An additional discussion about the temporal aspect of books and the success and non-success instances of the dataset can be visited in Section II of the S1 File.

Once the dataset was ready, we cleaned up the textual content of the 219 texts to maintain only the relevant contents of the books. In this process, the header and footer included by *Project Gutenberg* were removed, as well as editor/translator/author notes, captions and illustration indications, glossaries, footnotes, side-notes, annexes, and appendices. The dataset, in its final format, was made available at GitHub (https://github.com/giovanadanieles/bestSellersDataset).

## 5 Methodology

### 5.1 Text preprocessing

Using the dataset as explained in the previous section, the preprocessing of our analysis started. First, we replaced all capital letters with their corresponding lowercase counterparts. Then, the stopwords (i.e., words that provide low or no additional meaning to the context, such as articles and connectives) were removed. Next, we performed the tokenization of the books, in which elements, like punctuations and numbers, were disregarded. Finally, the obtained words were lemmatized—being lemmatization a technique whose objective is to reduce a vocable to its canonical form and to group different forms of the same word (e.g., the term "boys" is reduced to "boy" and "took" becomes "take"). Table 1 shows an example of this preprocessing.

### 5.2 Text embeddings

Techniques to embed textual content have been extensively used for a variety of tasks, including grasping text similarity, sentiment analysis, and classification. Among the most widely used techniques is the bag-of-words [12] approach, in which the relative frequencies of words appearing in a document are organized as a vector.

Recently, other approaches, now based on neural networks, have been developed to obtain dense embedding representations of words, sentences, or entire documents, being those approaches trained to predict masked parts in texts. In this sense, among the most used techniques is word2vec, which is based on a network comprising one hidden layer and a softmax output layer. The output layer is trained for predicting the context (words appearing together) given a focus word in a sentence [21]. For a given set of sentences, such a process provides an embedding for each word.

**Table 1. Preprocessing example.** Preprocessing of the excerpt "*It is difficult to live up to this kind of thing, and my thoughts drift to the auld schule-house and Domsie.*", obtained from the book *Beside the Bonnie Brier Bush*, by *Ian Maclaren*. In the column titled *Initial* is the original excerpt; in the next, the phrase without capital letters and stopwords; finally, in the last, the extract after tokenization and lemmatization processes.

| Initial | Removed capital letters and stopwords | Tokenized and lemmatized |
|---|---|---|
| *It is difficult* | *difficult* | ['*difficult*'] |
| *to live up to* | *live* | ['*live*'] |
| *this kind of* | *kind* | ['*kind*'] |
| *thing, and my* | *thing,* | ['*thing*'] |
| *thoughts drift* | *thoughts drift* | ['*thought*', '*drift*'] |
| *to the auld* | *auld* | ['*auld*'] |
| *schule-house* | *schule-house* | ['*schule*', *house*'] |
| *and Domsie.* | *domsie.* | ['*domsie*'] |

More sophisticated techniques such as BERT [22] and sentence BERT [23] generate embeddings that capture richer context and semantic information of words or sentences. However, these techniques, similar to W2V and GloVe [24], are limited to a small number of tokens and can not be applied to large portions of texts, such as entire books. For this reason, we opted to use the doc2vec (D2V) method to extract a vector representation of each book [13] since it has been successfully used in classification texts using large external corpora [25].

The doc2vec approach is based on the traditional word2vec [21] pipeline with the addition of the document tags as input. More specifically, it constitutes a neural network of three layers (*input*, *hidden*, and *softmax*), as illustrated in Fig 1a. Just like in word2vec with a continuous bag-of-words (CBOW) architecture, the inputs are one-hot vectors representing a sequence of words from a sentence in a book. A target word is omitted from the input and used to train the neural network. In addition, the input includes an extra one-hot vector identifying the book. The model is trained to predict the target word from the context (words adjacent to the target) using a negative sampling strategy. The vectors in the hidden layer connected directly to the books encoded as one-hot are used as the book embedding. Here, we opted to use the Gensim [26] software to obtain the doc2vec representations of books.

## 5.3 Visualization

Neural network embeddings usually result in high-dimensional dense vectors that are not correlated among themselves, which limits the use of linear techniques to reduce the dimensionality of these spaces (such as PCA [27]). Thus, the process of visualizing such structures is usually undertaken using non-linear projections, such as t-SNE [28] and UMAP [29].

However, embeddings can encode many different aspects of the data, for instance, a certain axis in a book embedding may be related to its number of pages or its adherence to the non-fiction or fantasy genres. The SemAxis approach [15] is a way to find an axis in a high-dimensional embedding that describes a certain aspect of the data. This is accomplished by first obtaining the centroids of two classes, e.g., small vs. larger books or non-fiction vs. fantasy

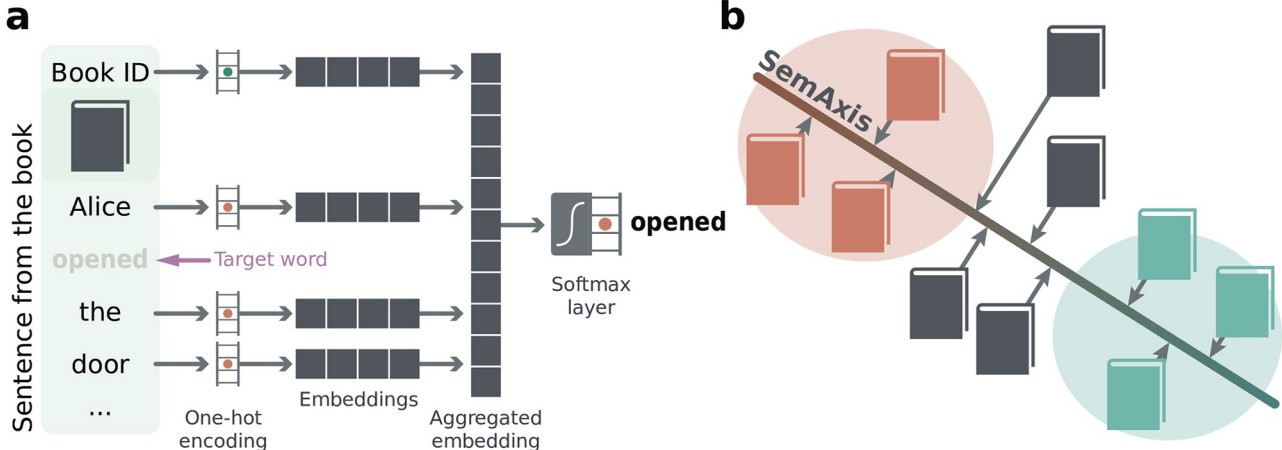

**Fig 1. Representation of doc2vec and SemAxis approaches.** In (a) we illustrate the neural network employed to obtain the embedding representation of books based on sequences of words (encoded as one-hot vectors) extracted from a book. The network is trained to predict a target word in the sentence based on the adjacent terms. Additionally, the original book ID is also encoded as input to the neural network, and their respective trained vectors correspond to the embedding space of books. In (b), we illustrate the SemAxis approach in which the line connecting the two categories' (*success* vs. *others*) centroids defines an axis to project all the books. This process results in a continuous one-dimensional (scalar) representation of books, which is employed for visualization purposes.

books. The line connecting the two centers define an axis in which all the remaining books are projected. This process is illustrated in Fig 1. Since in the current work we are interested in encoding the success of books, we employed the SemAxis approach to finding an axis for samples of the *success* and *other* classes. Similarly, in addition to SemAxis, we also employed linear discriminant analysis (LDA) [14], which also results in an axis encoding a continuous representation of the two classes.

In contrast to neural network-based approaches, the bag-of-words embedding can result in highly correlated and sparse vectors. For instance, the frequency patterns of two close-related words can correlate strongly and rare words may only be present in a small set of documents. Nonetheless, both LDA and SemAxis are still applicable in these conditions.

## 5.4 Classification: Distinguishing successes from others

The identification and classification of textual patterns were performed using traditional well-known machine learning classifiers [30]. We considered different classifier strategies, including *k*-nearest neighbors (KNN) [31] (based on the probable similarity of nearest neighbors), naive Bayes (NB) [32] (that estimates the class-conditional probability based on the Bayes theorem and assuming conditional independence between attributes), decision tree (DT) [33] (which classifies an example of the test record based on a series of discriminating questions about its attributes), support-vector machine (SVM) [30, 34] (based on finding hyper-planes that can linearly separate data—called support vectors), and, finally, the two that yielded the best results: random forest (RF) [35] and logistic regression (LR) [36].

In just a few words, Random Forest is a class of ensemble methods designed over DT classifiers. It uses multiple decision trees, built using a set of random vectors, combining each of their predictions to yield a final classification. On the other hand, Logistic Regression is based on determining the conditional probability of an event happening. It models this probability by minimizing a negative likelihood function for the labeled classes.

All these tests were implemented in Python language [37] using the classifiers of Scikit-Learn [38] library. Following the guidelines described in related works [39, 40], we used the default parameters of the methods to classify texts. As an exception, in the case of the SVM, we changed the parameter "max_iter", the maximum number of iterations, to 10,000.

## 6 Results and discussions

This section describes the experiments performed to study the task of automatically characterizing and identifying best seller books. The proposed data analysis pipeline is illustrated in Fig 2. First, we obtain word embedding representations of each book by employing two distinct techniques (Fig 2a): bag-of-words and doc2vec (the latter with different dimensions, namely 32, 64, 128, and 256). Next, we investigate the proposed classification problem through two main approaches: visualization and classification. In the first, we employed a simple visualization pipeline to verify and illustrate the potential of using embeddings to identify best seller books (Fig 2b–2d). The objective of this approach is to provide a preliminary and simple way to visually inspect the considered high-dimensional embeddings by summarizing them into a single continuous axis.

The visualization pipeline starts with the standardization of the obtained embeddings (Fig 2b). To reduce the dimensionality of the embeddings, we employed SemAxis [15] and LDA [14]. Since these methods are supervised, the final visualizations are performed by employing the leave-one-out technique to avoid overfitting.

The second approach considered in this work is the direct application of classification methods, allowing quantitative comparison of the respective performance. For that, we

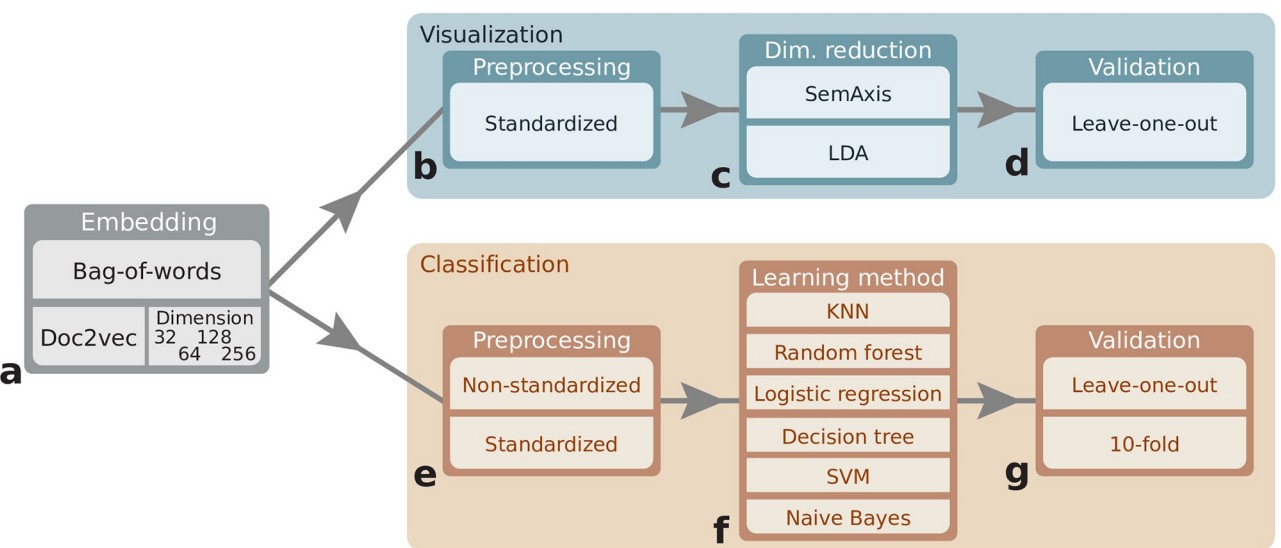

**Fig 2. Overall diagram of the main approaches.** All the methods within the blue and orange boxes are applied to the two considered embeddings in a combined fashion. For example, a valid path would be: (i) embedding: doc2vec with dimension equals to 32; (ii) preprocessing: standardized; (iii) learning method: logistic regression; (iv) validation: leave-one-out.

employed a pipeline comprising the same embedding configurations as before but followed by three successive stages: preprocessing, learning method, and validation, each presented as a box in Fig 2. All combinations between the components of each of these boxes are considered in our evaluation.

In this sense, the following two first subsections are intended to detail the task of visualization, followed by the classification, both using the bag-of-words and then the doc2vec representation. Then, in the last subsection, we repeat these experiments to evaluate a specific variation of the constructed dataset. Additionally, for those interested in results using non-full-text content, there is an additional discussion in Section I of the S1 File. In it, we explored only the beginning of each book. Moreover, we also discuss using readability measures and textual features to discriminate between bestsellers and non-bestsellers in Section III of the S1 File.

### 6.1 Bag-of-words analysis

The first performed experiment intends to evaluate whether the frequency of words composing the books can discriminate between best sellers and ordinary literary works. For this purpose, we considered the set $S$, built based on the 3,585 different words that appeared at least in $\frac{N}{2}$ texts of the dataset. The proportion $\frac{N}{2}$ was elected once smaller ones (such as $\frac{N}{3}$ or $\frac{N}{4}$, being $N$ the total number of books in the dataset) evoked archaic words and words not belonging to the vernacular of the English language, and higher proportions, on the contrary, led to poorer results on the experiments.

Considering each entry in $S$, we computed its frequency for all books in the dataset, resulting, in the end, in a $219 \times 3585$ matrix of frequencies, henceforward called $M$. Next, the rows of $M$—each representing a book—were standardized and transformed according to two approaches: LDA and SemAxis, the results being cross-validated through leave-one-out. As shown in Fig 3, such processing led to a visual separation both in **a** and **b**, giving evidence that the bag-of-words model can provide a good—although not exact—split between the two studied categories.

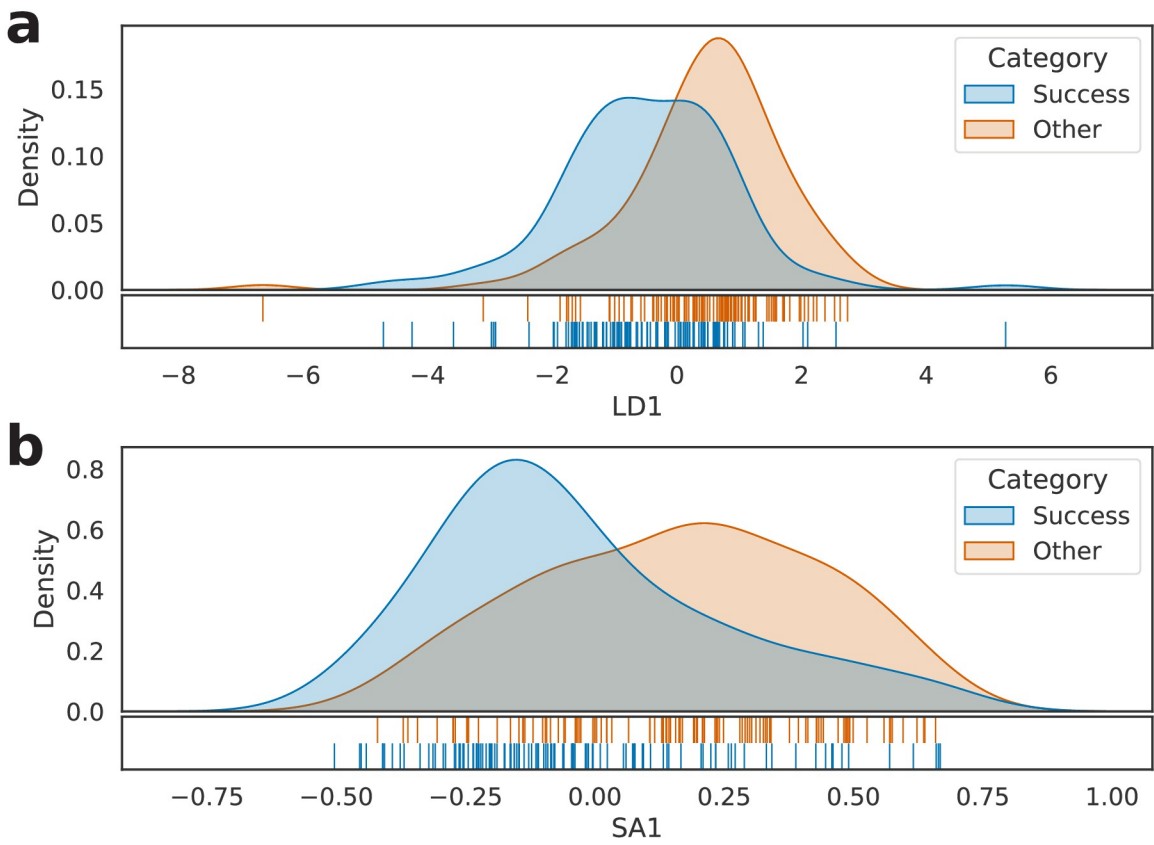

**Fig 3. Kernel density estimation of the 219 investigated literary works.** (a) LDA projection and (b) SemAxis projection of *M*.

Moreover, to quantitatively assess the obtained separation, *M* was used as input to supervised classification methods (videlicet: KNN, logistic regression, naive Bayes, decision tree, random forest, and SVM). We also applied leave-one-out and *k*-fold (taking *k* = 10) cross-validation methods and considered both the standardized and the non-standardized versions of *M* (the standardized version denoted by $\hat{M}$). Here, we adopted the standard hyperparameters of each model—in other words, not involving tuning operations.

As shown in Table 2, the linear regression model resulted as the best choice for grasping discrepancies between classes, leading to an average classification accuracy of 0.75, either for

**Table 2. Classification accuracy for different models and arrangements.** Results for configurations *M* or $\hat{M}$ and leave-one-out or *k*-fold cross-validation. Highlighted in bold is the best result for each configuration.

| | *M* | | $\hat{M}$ | |
|---|---|---|---|---|
| | **LOO** | **10-fold** | **LOO** | **10-fold** |
| foKNN | 0.64 | 0.64 ± 0.11 | 0.58 | 0.56 ± 0.10 |
| LR | 0.65 | 0.64 ± 0.13 | **0.75** | **0.75 ± 0.09** |
| NB | 0.63 | 0.62 ± 0.11 | 0.63 | 0.62 ± 0.11 |
| DT | 0.65 | 0.58 ± 0.14 | 0.65 | 0.58 ± 0.14 |
| RF | **0.68** | **0.68 ± 0.11** | 0.68 | 0.68 ± 0.11 |
| SVM | 0.66 | 0.63 ± 0.11 | 0.72 | 0.74 ± 0.09 |

Leave One Out (LOO) or $k$-fold cross-validation. This result shows that the approach is apt—to a reasonable extent—to identify successful literary works. Furthermore, it is worth observing that the standardization positively impacts the outcomes, leading to performances as good as or better than the non-standardized case in ten out of twelve scenarios—languishing only the accuracy of the KNN model. Please consult Section IV of the S1 File for complementary information concerning precision, recall, and f1-score metrics results.

In addition, we retrieved the 40 words of $S$ with preponderant impact onto the SemAxis projection, aiming at analyzing what sort of vocable seems to be characteristic in best sellers and in the *other* books. As presented in Table 3, the most meaningful words for successful books encompass six adjectives, nine nouns, one adverb, and four verbs; for the non-best seller books, we have three adjectives, seven nouns, one adverb, and nine verbs. Similarly to a result formerly reported in [19], words referring to body parts (such as *eye*, *face*, and *hand*) play a central role in less successful titles. Furthermore, none of the 20 most relevant terms for successes ranks among the 40 most frequent words of $S$—however, when analyzing the non-best seller books, the principal words *eye*, *face*, *hand*, and *back* represent, respectively, the 5th, 6th, 10th, and 12th most common words of the dataset.

## 6.2 Doc2vec analysis

The second experiment evaluates whether doc2vec's representation of literary works can grasp the dissimilarity between the two analyzed classes. With this aim, we instantiated D2V models with 32, 64, 128, and 256 dimensions (a feature commonly called vector size, hereafter referred to as $\#_2D$). We also set the minimum word count to 1 (to ignore all words with a total frequency lower than one), the window (maximum distance between the current and predicted term within a sentence) to 5, and the epochs (number of iterations over the corpus) to 40. Lastly, the model training occurred using all 219 instances of the dataset.

Next, each model vector (henceforth called $D$)—a piece representing a different book—was transformed employing LDA and SemAxis techniques along with leave-one-out cross-validation, yielding the results shown in Fig 4. As can be observed, the method was able to characterize best seller and non-best seller works in a contrasting fashion, both in **a** and **b**. This result shows that it is possible to emphasize the differences between the two classes in two noticeably distinct approaches (either BoW or D2V).

**Table 3. Forty most significant words to the SemAxis projection discrimination between best sellers and others.** The importance of each term for the method dictates its allocation order in the table: the element on the first row and the first column (of success/other) is the most important for the class; the one on the second row and the second column is the second most important; and so forth.

| | Vocables | | | |
|---|---|---|---|---|
| Success | *ordinary* | *evidence* | *motive* | *exhibit* |
| | *grey* | *substance* | *improve* | *copy* |
| | *instruction* | *contain* | *examination* | *practice* |
| | *accordingly* | *teacher* | *numerous* | *interesting* |
| | *school* | *large* | *attach* | *average* |
| Other | *breath* | *drop* | *draw* | *sharply* |
| | *face* | *eye* | *turn* | *stun* |
| | *break* | *hand* | *push* | *back* |
| | *reckless* | *caught* | *arm* | *shake* |
| | *tone* | *instant* | *quick* | *glance* |

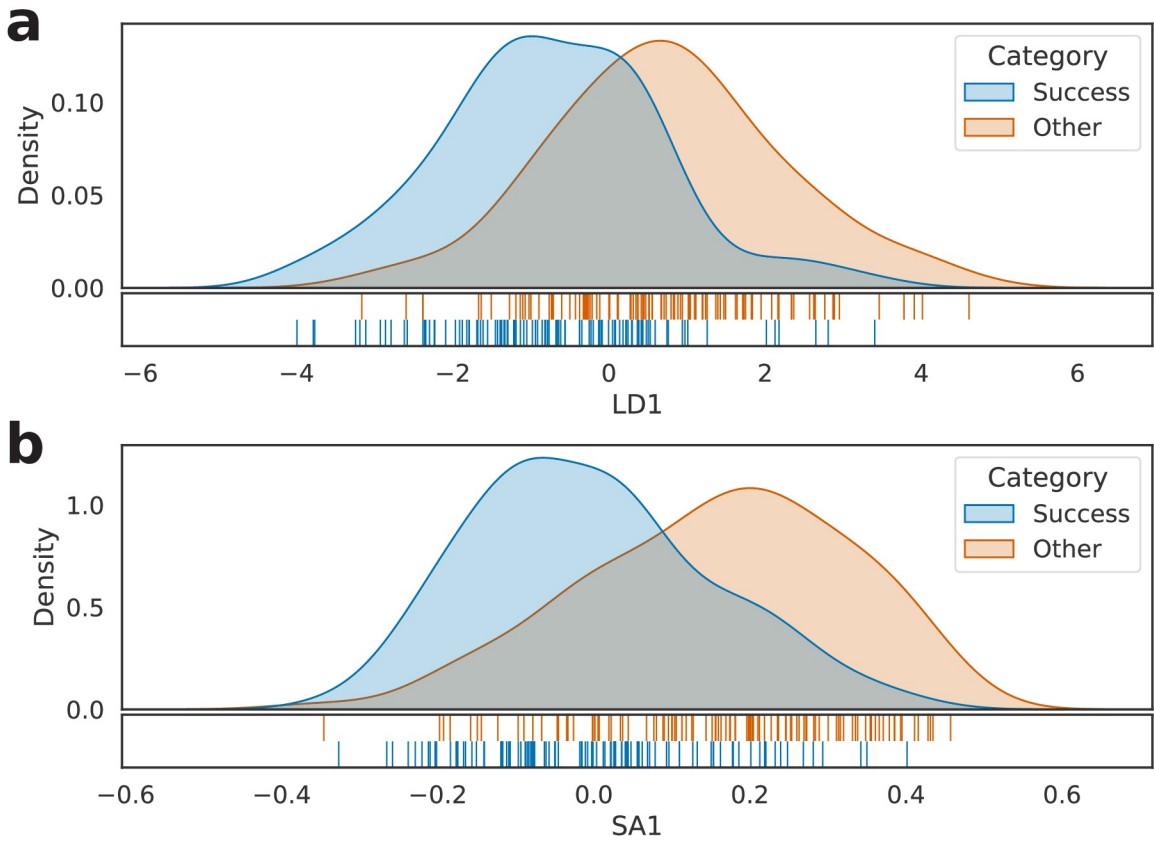

**Fig 4. Kernel density estimation of the 219 investigated literary works.** (a) LDA projection and (b) SemAxis projection of D2V representation (adopting $\#_2 D = 64$).

Furthermore, we used $D$ as the supervised classification methods' input to quantitatively assess the obtained separation. The models used here were the same as those applied in the BoW experiment, and we also considered LOO and 10-fold cross-validations and both the standardized and the non-standardized versions of $D$ (the standardized version denoted by $\hat{D}$). The chosen models' hyperparameters were the standard ones.

As shown in Tables 4 and 5, naive Bayes was the model that best performed the task of distinguishing classes considering the D2V representation, leading to a maximum classification

**Table 4. Classification accuracy for LOO cross-validation combined with different models and arrangements.** Results for configurations $D$ or $\hat{D}$ and for D2V vector size: 32, 64, 128, or 256. Highlighted in bold is the best result for each configuration.

| | LOO | | | | | | | |
| --- | --- | --- | --- | --- | --- | --- | --- | --- |
| | $D$ | | | | $\hat{D}$ | | | |
| | **32** | **64** | **128** | **256** | **32** | **64** | **128** | **256** |
| KNN | 0.65 | 0.65 | 0.65 | 0.67 | 0.65 | 0.65 | 0.64 | 0.67 |
| LR | 0.68 | 0.65 | 0.63 | 0.66 | 0.70 | 0.64 | 0.61 | 0.67 |
| NB | **0.71** | **0.68** | **0.69** | **0.68** | **0.71** | **0.68** | **0.69** | **0.68** |
| DT | 0.48 | 0.58 | 0.53 | 0.40 | 0.48 | 0.58 | 0.53 | 0.40 |
| RF | 0.66 | 0.63 | 0.67 | 0.64 | 0.66 | 0.63 | 0.67 | 0.64 |
| SVM | 0.68 | 0.63 | 0.62 | 0.63 | 0.69 | 0.66 | 0.65 | 0.64 |

**Table 5. Classification accuracy for 10-fold cross-validation combined with different models and arrangements.** Results for configurations $D$ or $\hat{D}$ and for D2V vector size: 32, 64, 128, or 256). Highlighted in bold is the best result for each configuration.

| | 10-fold | | | |
| --- | --- | --- | --- | --- |
| | $D$ | | | |
| | 32 | 64 | 128 | 256 |
| KNN | 0.67 ± 0.08 | 0.63 ± 0.06 | 0.65 ± 0.13 | 0.69 ± 0.11 |
| LR | 0.66 ± 0.13 | 0.67 ± 0.10 | 0.59 ± 0.08 | 0.71 ± 0.09 |
| NB | **0.68 ± 0.10** | **0.70 ± 0.13** | **0.70 ± 0.08** | **0.72 ± 0.12** |
| DT | 0.58 ± 0.12 | 0.52 ± 0.08 | 0.56 ± 0.10 | 0.50 ± 0.06 |
| RF | **0.68 ± 0.10** | 0.65 ± 0.10 | 0.66 ± 0.09 | 0.70 ± 0.10 |
| SVM | 0.66 ± 0.13 | 0.66 ± 0.09 | 0.57 ± 0.07 | 0.69 ± 0.10 |
| | $\hat{D}$ | | | |
| | 32 | 64 | 128 | 256 |
| KNN | **0.68 ± 0.11** | 0.64 ± 0.05 | 0.66 ± 0.10 | 0.68 ± 0.08 |
| LR | **0.68 ± 0.11** | 0.69 ± 0.12 | 0.60 ± 0.08 | 0.70 ± 0.10 |
| NB | **0.68 ± 0.10** | **0.70 ± 0.13** | **0.70 ± 0.08** | **0.72 ± 0.12** |
| DT | 0.58 ± 0.12 | 0.52 ± 0.08 | 0.56 ± 0.10 | 0.50 ± 0.06 |
| RF | **0.68 ± 0.10** | 0.65 ± 0.10 | 0.66 ± 0.09 | 0.70 ± 0.10 |
| SVM | 0.67 ± 0.14 | 0.64 ± 0.11 | 0.57 ± 0.08 | 0.70 ± 0.10 |

accuracy of 0.71 for LOO and 0.72±0.12 for 10-fold. In the LOO version, $\#_2 D = 32$ raises the best results, while $\#_2 D = 256$ performs better for 10-fold. Although the standardization did not affect the naive Bayes classifier, it led to the same or slightly better outcomes for the others—the exceptions being some arrangements, namely KNN (for $\#_2 D = 128$) and LR (for $\#_2 D = 64$ and 128) for the LOO version and KNN ($\#_2 D = 256$), LR ($\#_2 D = 256$), and SVM ($\#_2 D = 64$) for 10-fold.

### 6.3 Are the subjects being grasped by the approaches?

What if the above-mentioned approaches are only grasping (or being biased by) the subjects of the books? That would be a valid inquiry once we did not regard this type of information during the construction of the database. To assess this possibility, we retrieved the list of subjects of each book provided by the Gutenberg platform and then analyzed the ten most common ones in the dataset. In Fig 5, we plot those subjects against the SemAxis projection of the books' D2V vector representation (using $\#_2 D = 64$), stratifying the results by category. As one can see, the only subjects with a representative number of instances are PS and PR, which also seem to explain the separation obtained through the D2V method to some degree.

PR and PS are classifications used by the Library of Congress [41] to catalog English and British literature, respectively. In our case, the PR subject represents 102 instances of the dataset, 34 best sellers, and 68 non-best seller works. The PS one, by contrast, encompasses 98 books—72 best sellers and 26 other types of works. In this manner, as, in principle, the success category is the only one with a limited number of instances (given its criteria), we created a new dataset (with 72 successes and 72 others, embracing the same standards stated in the creation of the former dataset) with only literary works belonging to subject PS. Then, 46 new non-best seller titles were selected from the Gutenberg platform. Using this current dataset, we repeated the previous experiments, aiming at understanding whether the fact that a book belongs to English or British literature was enough to explain the separation provided by the BoW and D2V methods. The results are presented and discussed below.

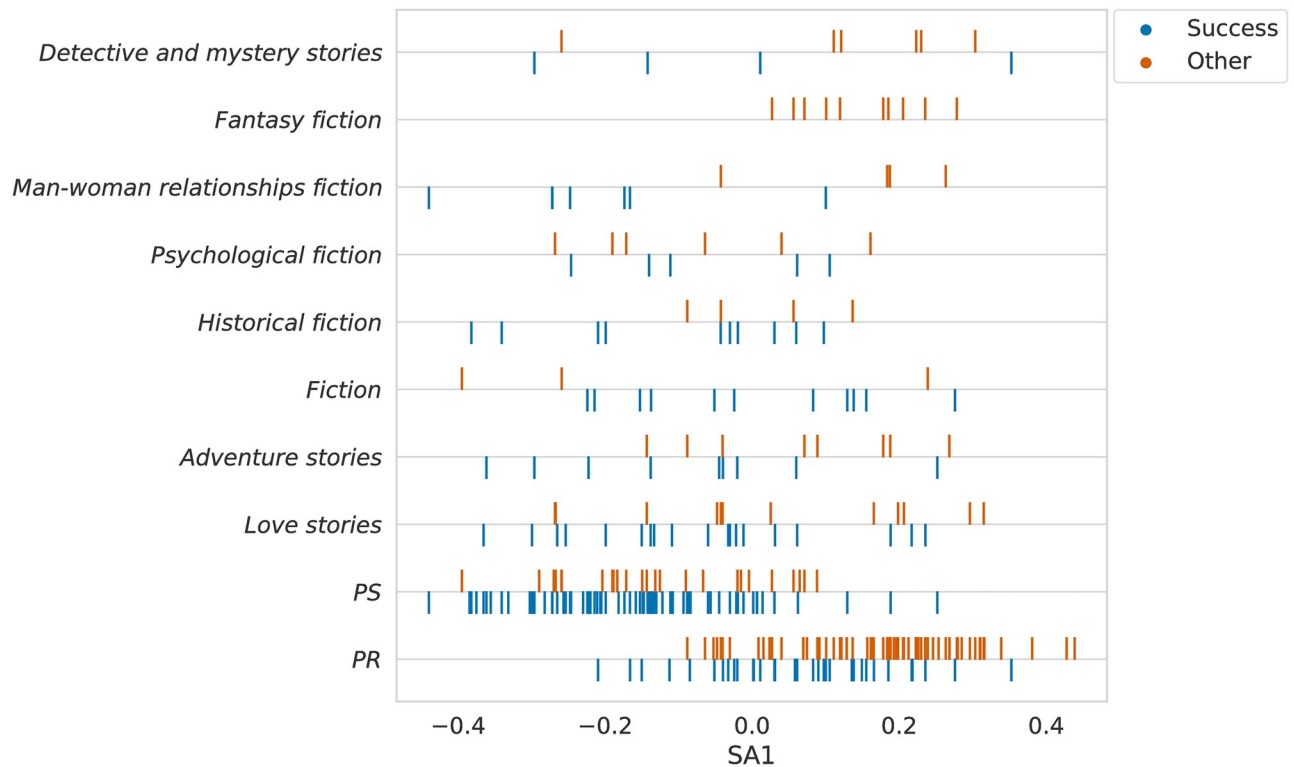

**Fig 5. On the y-axis, the ten most common subjects in the dataset.** On the x-axis, the SemAxis projection of the books' D2V representation (adopting $\#_2D = 64$).

**6.3.1 Bag-of-words analysis.** Similarly to the previous experiment, we considered the set $S$, built based on the 3,257 different words that appeared in the text of at least $\frac{N}{2}$ books of the PS dataset. Then, we calculated their frequencies, resulting in a $144 \times 3257$ dimension matrix (henceforth called $M_{PS}$). Next, $M_{PS}$'s rows were standardized, transformed using LDA and SemAxis, and verified via LOO cross-validation, as shown in Fig 6. It is possible to observe that the separation between the classes is still perceptible—both in **a** and **b**—although now solely English literature is being considered.

To quantitatively analyze the separation of the categories, we performed supervised classification methods using the standardized and non-standardized versions of $M_{PS}$ as input. The applied models, hyperparameters, and cross-validation methods were the same as the former experimentation. As shown in Table 6, the random forest model was the best option to distinguish the best seller and other instances, leading to an average accuracy of 0.71—both in LOO and 10-fold cross-validations. The standardization did not affect the results. Even though this is a lower accuracy than that obtained with the previous dataset (which led to the highest accuracy of 0.75), it is worth mentioning that the PS dataset has 35% fewer instances than the other, which makes us expect lower accuracies and higher standard deviations. Thus, it is possible to state that the BoW method is not classifying the corpus instances based predominantly or solely on their literary class.

**6.3.2 Doc2vec analysis.** The D2V models were instantiated in the context of the new dataset with the same hyperparameters as the previous tests—the only exception being that we trained the model using 144 books instead of 219. We repeated the former configurations

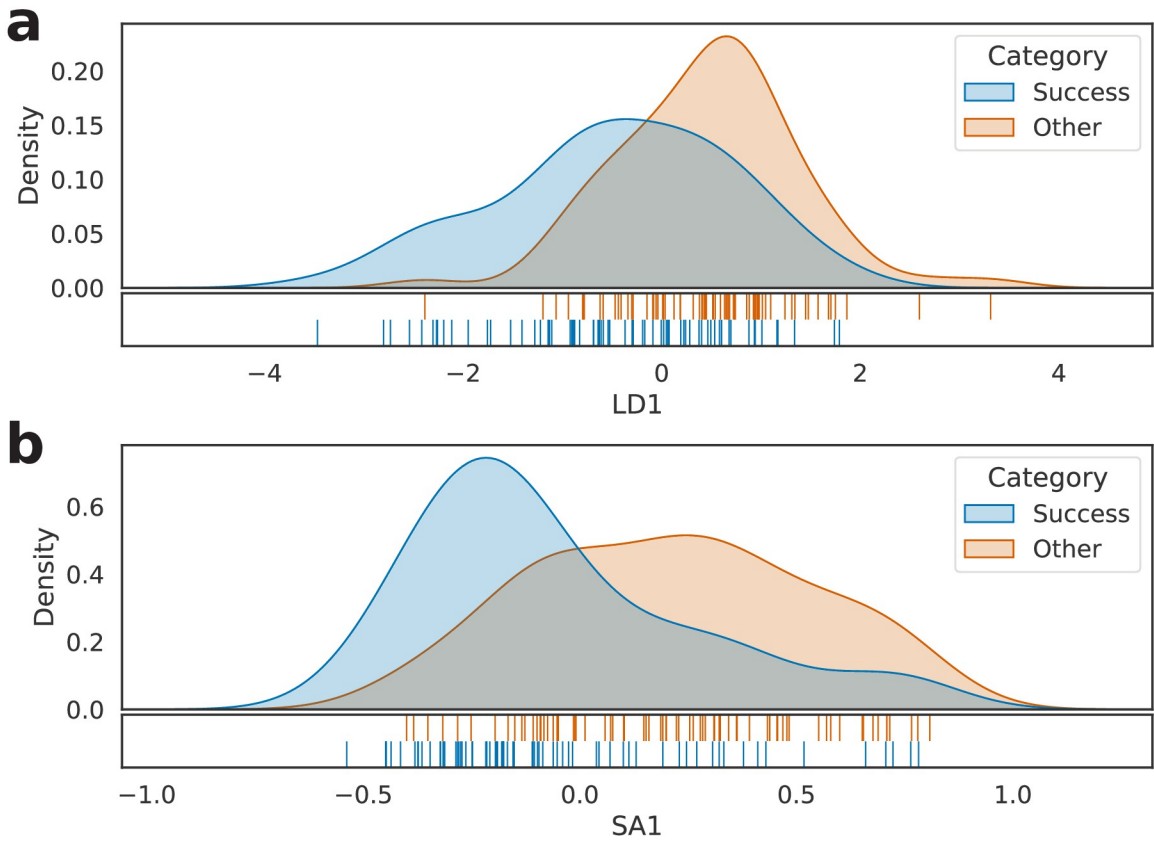

**Fig 6. Kernel density estimation of the 144 investigated literary works belonging to the PS subject.** (a) LDA projection and (b) SemAxis projection of $M_{PS}$.

(vector sizes 32, 64, 128, and 256 and LOO and 10-fold cross-validations) and adopted the standardized and the non-standardized versions of the model vectors—called $\hat{D}_{PS}$ and $D_{PS}$, respectively. Fig 7 shows the results of the transformations of the model vectors via LDA and SemAxis. The split between best seller and non-best seller works was again observed, suggesting that the method is insensitive to the literary class.

The quantitative assessment using supervised classification led to the results shown in Tables 7 and 8. From Table 8, it is possible to conclude that the models that best performed for

**Table 6. Classification accuracy for different models and arrangements considering the PS dataset.** Results for configurations $M_{PS}$ or $\hat{M}_{PS}$ and leave-one-out or $k$-fold cross-validation.

| | $M_{PS}$ | | $\hat{M}_{PS}$ | |
|---|---|---|---|---|
| | **LOO** | **10-fold** | **LOO** | **10-fold** |
| KNN | 0.62 | 0.60±0.09 | 0.63 | 0.62±0.13 |
| LR | 0.64 | 0.63±0.15 | 0.69 | 0.67±0.13 |
| NB | 0.65 | 0.65±0.15 | 0.65 | 0.65±0.15 |
| DT | 0.62 | 0.57±0.14 | 0.62 | 0.57±0.14 |
| RF | **0.71** | **0.71±0.14** | **0.71** | **0.70±0.12** |
| SVM | 0.61 | 0.61±0.16 | 0.66 | 0.61±0.16 |

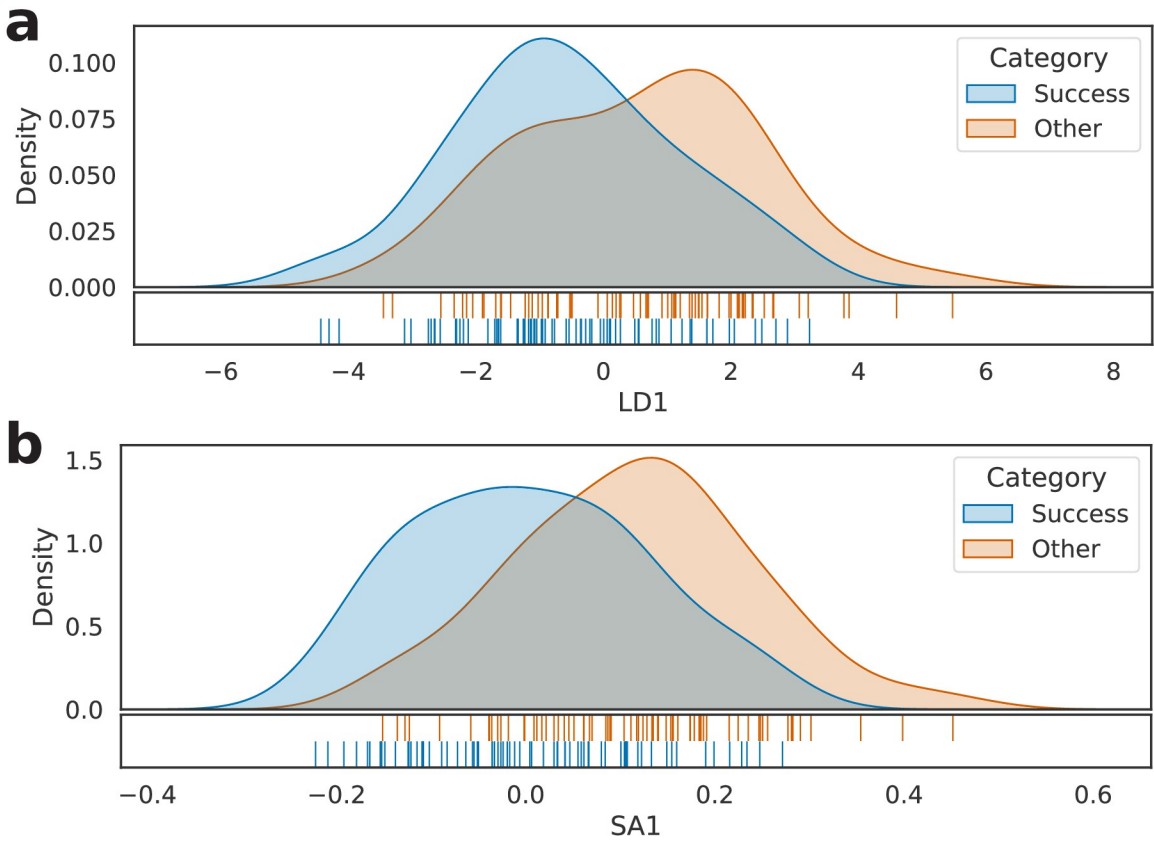

**Fig 7. Kernel density estimation of the 144 investigated literary works belonging to the PS subject for the D2V representation.** (a) LDA projection and (b) SemAxis. In both cases, $\#_2 D_{PS} = 64$ was adopted.

LOO cross-validation were logistic regression and naive Bayes—the highest accuracy (0.67) given by the latter, with $\#_2 D_{PS} = 256$. In this case, the standardization did not contribute to performance improvement only in the case of the KNN model. From Table 8, we conclude that no model stood out for 10-fold, with the highest accuracy of 0.67 given by the SVM model, with $\#_2 D_{PS} = 32$ and non-standardized input. The standardization process induced better

**Table 7. Classification accuracy for LOO cross-validation combined with different models and arrangements (i.e., whether $D_{PS}$ or $\hat{D}_{PS}$ were employed and with which D2V vector size: 32, 64, 128, or 256), considering the PS dataset.** Highlighted in bold is the best result for each configuration.

| | LOO | | | | | | | |
|---|---|---|---|---|---|---|---|---|
| | $D_{PS}$ | | | | $\hat{D}_{PS}$ | | | |
| | 32 | 64 | 128 | 256 | 32 | 64 | 128 | 256 |
| KNN | 0.56 | 0.59 | 0.60 | 0.56 | 0.51 | 0.57 | 0.56 | 0.55 |
| LR | **0.62** | **0.62** | 0.58 | 0.60 | **0.62** | **0.65** | 0.60 | 0.60 |
| NB | 0.60 | **0.62** | **0.62** | **0.67** | 0.60 | 0.62 | **0.62** | **0.67** |
| DT | 0.51 | 0.51 | 0.44 | 0.53 | 0.51 | 0.51 | 0.44 | 0.53 |
| RF | 0.53 | 0.60 | 0.58 | 0.56 | 0.53 | 0.60 | 0.58 | 0.56 |
| SVM | **0.62** | 0.61 | 0.60 | 0.58 | **0.62** | 0.61 | **0.62** | 0.58 |

**Table 8. Classification accuracy for 10-fold cross-validation combined with different models and arrangements (i.e., whether $D_{PS}$ or $\hat{D}_{PS}$ were employed and with which D2V vector size: 32, 64, 128, or 256), considering the PS dataset.** Highlighted in bold is the best result for each configuration.

| | 10-fold | | | |
|---|---|---|---|---|
| | $D_{PS}$ | | | |
| | 32 | 64 | 128 | 256 |
| KNN | 0.59 ± 0.13 | 0.58 ± 0.14 | 0.49 ± 0.12 | 0.60 ± 0.09 |
| LR | **0.66 ± 0.13** | 0.54 ± 0.06 | 0.59 ± 0.12 | **0.66 ± 0.08** |
| NB | 0.61 ± 0.08 | 0.59 ± 0.12 | 0.58 ± 0.14 | 0.61 ± 0.14 |
| DT | 0.45 ± 0.13 | 0.49 ± 0.10 | **0.66 ± 0.13** | 0.48 ± 0.09 |
| RF | 0.59 ± 0.10 | **0.60 ± 0.10** | 0.57 ± 0.17 | 0.59 ± 0.13 |
| SVM | 0.65 ± 0.12 | 0.49 ± 0.10 | 0.58 ± 0.12 | 0.63 ± 0.10 |
| | $\hat{D}_{PS}$ | | | |
| | 32 | 64 | 128 | 256 |
| KNN | 0.60 ± 0.13 | **0.61 ± 0.16** | 0.53 ± 0.07 | 0.59 ± 0.08 |
| LR | 0.66 ± 0.12 | 0.58 ± 0.09 | 0.59 ± 0.12 | **0.65 ± 0.09** |
| NB | 0.61 ± 0.08 | 0.59 ± 0.12 | 0.58 ± 0.14 | 0.61 ± 0.14 |
| DT | 0.45 ± 0.13 | 0.49 ± 0.10 | **0.66 ± 0.13** | 0.48 ± 0.09 |
| RF | 0.59 ± 0.10 | 0.60 ± 0.10 | 0.57 ± 0.17 | 0.59 ± 0.13 |
| SVM | **0.67 ± 0.11** | 0.53 ± 0.07 | 0.55 ± 0.14 | 0.63 ± 0.10 |

results on six distinct occasions, although the best-obtained accuracy counts on a non-standardized vector.

For the 219-instances dataset, the best-achieved accuracy was 0.72. Again, we expected a drop in the accuracy, as the new dataset has 35% lesser instances than the other. Thus, just as in the BoW method, it is possible to infer that the separation between classes obtained in the D2V approach does not rely solely on whether a book belongs to English or British literature.

## 7 Conclusions

The study of characteristics leading to literary pieces becoming best sellers constitutes an intriguing and challenging research subject. The present work addressed this issue while considering aspects derived from the full content of a list of more and less successful books retrieved from the Gutenberg Project, based on the best seller lists of Publishers Weekly. Several alternative content representation, standardization, visualization, and classification approaches were considered, as summarized in the diagram shown in Fig 2.

We started our analysis by examining the data using visualization techniques. The visualization enabled a preliminary direct inspection of the embedding by looking at a single axis that maximizes the separation between best sellers and ordinary books. Specifically, we employed SemAxis and LDA techniques—the first providing better discrimination between classes than the latter, both for bag-of-words and doc2vec representations. Furthermore, SemAxis provided means that helped to: (i) understand the most characteristic words in best sellers and non-best sellers; and (ii) check if the respective success was related to the subjects of the books (e.g., love stories, adventure stories, fiction, among others). In line with earlier work [19], words related to body parts (like *face*, *eye*, and *hand*) played a central role in non-best seller books, while more varied and less common vocables (such as *ordinary*, *accordingly*, and *examination*) were characteristic of more successful books. Moreover, we found no evidence that the subject of the books impacted the class discrimination obtained.

For the classification tasks, we tested two strategies for preprocessing the two distinct representations: (i) standardizing and (ii) non-standardizing the embeddings. Then, we evaluated the proposed representations via different classifiers (namely: KNN, LR, NB, DT, RF, and SVM). The best-obtained result was acquired with the complete dataset (219 books) using the LR classifier with the standardized bag-of-words representation. In this case, the final classification accuracy was 0.75. Still dealing with the complete set, the best accuracy obtained for D2V embedding was 0.72, combining the standardized representation with the NB model. For the dataset considering only the PS subject (144 books), the bag-of-words approach throughput the most promising results for the standardized data inputted in the RF classifier. The D2V representation, in contrast, returned better outcomes for the standardized data combined with the NB classifier. These results agree with the tendency of the two classes' separation found in the visualization analysis. Interestingly, the standardization did not affect the results significantly in the doc2vec approach for both datasets.

The reported methodology and results pave the way for several related studies, some of which are described as follows. Firstly, it would be interesting to adapt the reported method to other types of embeddings, for example, the BERT transformer modified to work with long texts. Secondly, it would be interesting to consider the described approach for better understanding: (i) other types of documents, such as scientific books and articles, and (ii) additional types of artistic production, including music, poetry, and theater. Lastly, another point that could be explored concerns the explanation of additional reasons why some literary works become best sellers and others do not.

Concerning the limitations of the work, the three main points we stress are (i) the absence of modern books in the database, (ii) the absence of more modern modeling techniques, and (iii) the limitation in dataset size imposed by the number of available best-selling books. As previously discussed, the scarcity of books is due to copyright laws that protect the complete contents of modern books. Even though such content is found free of charge on the internet, we do not have the right to use it. Regarding modeling, more modern techniques, such as BERT, do not deal well with long texts [42]. As the median size of our dataset is approximately 90,000 characters, it would not be appropriate to apply such a technique. Even if the modeling yielded highly accurate results, they would not be reliable. Finally, concerning the limited size of the dataset, we are restricted by the number of books listed as best sellers and also available in the public domain. As previously stated, we can not leverage books that don't have free content. Also, best-selling books are scarce per nature: if all books were best-selling pieces, this study would not even exist. In addition to these points, it is also worth mentioning that although an accuracy greater than 80 or 90% would be desirable, it would be unrealistic to predict the success of works with such a high outcome, given the intricate and multifaceted nature of such a task. Factors like marketing and trends can influence the popularity of books in ways difficult to predict or measure. Therefore, the 75% accuracy result becomes reasonable, although somewhat limited, if we think we are exploring solely the textual content of each book. Future models, incorporating additional factors such as marketing, author popularity and contextual elements, could offer a more comprehensive understanding of what drives a book's success.

Regardless of the dataset limitations we recognized in this research, we have employed careful effort to minimize any undesirable impact of external factors, such as authorship, publication period, and literary genre, being the book's full text the main feature considered in the hypothesis tested in this work. Ultimately, we were able to provide valuable insights into the factors that seem to lead to the relative success of a book in becoming a best seller, namely, the content of the text.

## Supporting information

**S1 File.**
(PDF)

## Author Contributions

**Conceptualization:** Giovana D. da Silva, Filipi N. Silva, Henrique F. de Arruda, Bárbara C. e Souza, Luciano da F. Costa, Diego R. Amancio.

**Data curation:** Giovana D. da Silva.

**Formal analysis:** Giovana D. da Silva.

**Funding acquisition:** Diego R. Amancio.

**Investigation:** Giovana D. da Silva, Filipi N. Silva, Henrique F. de Arruda, Luciano da F. Costa, Diego R. Amancio.

**Methodology:** Giovana D. da Silva, Filipi N. Silva, Henrique F. de Arruda, Luciano da F. Costa, Diego R. Amancio.

**Software:** Giovana D. da Silva.

**Supervision:** Filipi N. Silva, Henrique F. de Arruda, Luciano da F. Costa, Diego R. Amancio.

**Validation:** Giovana D. da Silva.

**Visualization:** Giovana D. da Silva.

**Writing – original draft:** Giovana D. da Silva.

**Writing – review & editing:** Giovana D. da Silva, Filipi N. Silva, Henrique F. de Arruda, Bárbara C. e Souza, Luciano da F. Costa, Diego R. Amancio.

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
