## [Decision Letter · Decision Letter 0]

10 May 2023

PONE-D-23-04137Using Full-Text Content to Characterize and Identify Best Seller BooksPLOS ONE

Dear Dr. Silva,

Thank you for submitting your manuscript to PLOS ONE. After careful consideration, we feel that it has merit but does not fully meet PLOS ONE’s publication criteria as it currently stands. Therefore, we invite you to submit a revised version of the manuscript that addresses the points raised during the review process.

We look forward to receiving your revised manuscript.

Kind regards,

Heba El-Fiqi

Academic Editor

PLOS ONE

Journal Requirements:

Reviewers' comments:

Reviewer's Responses to Questions

**Comments to the Author**

1. Is the manuscript technically sound, and do the data support the conclusions?

Reviewer #1: Partly

Reviewer #2: Partly

2. Has the statistical analysis been performed appropriately and rigorously? 

Reviewer #1: Yes

Reviewer #2: No

3. Have the authors made all data underlying the findings in their manuscript fully available?

Reviewer #1: Yes

Reviewer #2: Yes

4. Is the manuscript presented in an intelligible fashion and written in standard English?

Reviewer #1: Yes

Reviewer #2: Yes

5. Review Comments to the Author

Reviewer #1: Review

I think this article is basically an interesting study. The “mathematical side” of the argument is sound, but crucial aspects of the study are flawed. The dataset is not adequate for the conclusions that are drawn. The main approach should be better motivated. Its limitations should be discussed in deeper theoretical detail, and the conclusions should be stated with more caution.

The authors state: “This study aims to probe whether the full-text content of the book alone can indicate if it will become a best seller.” That would require surveying all possible methods, which isn’t feasible.

Same point again: “The obtained results suggest that it is infeasible to predict the success of a literary work with high accuracy by using only its full-text content.” But that conclusion cannot be drawn. A study of this kind can only show that the approaches that have been tested fail.

The authors rightly say that “the main trends, interests, and expectations predominating in a given period” are crucial for bestsellerhood. This theoretical argument is a stronger one than the empirical one proposed in the paper. The empirical results are however compatible with that theoretical statement. Important factors are text-external.

The conclusion “our experiments evince that the subject of the books does not seem to be a core factor for a title becoming a best seller” seems to be carelessly stated. It is hard to deny “that the subject of the books IS A CRUCIAL FACTOR for a title becoming a best seller”. The experiments cannot disprove that. I actually find the accuracy of 0.75 surprisingly high. It would rather support the conclusion that content is very important, had the data behind it been more convincingly curated.

The data selection is of the convenience kind. It is unclear how it would be representative of some relevant populations. The Gutenberg repository is hardly a representative selection of books. It is likely to have books interesting by their quality and/or popularity. Books belonging to typical bestseller genres which have failed to become bestsellers, say an unsuccessful crime novel, tend to be forgotten, and they are less likely to find their way to Gutenberg. If, as the authors correctly state, “the main trends, interests, and expectations” are important, bestsellers should be compared to “fail-sellers” in the same time and place aligning with the same trends, interests, and expectations. The size of the dataset should also be motivated.

So, “the other [class set] would have the same number of titles published in the same year — the titles randomly selected from the Gutenberg repository”. There are other relevant parameters to consider, mainly genre. But the “other” books include both literary works and non-fiction, whereas, I guess, the bestsellers to a high extent comprise genre fiction. (“the PS dataset” is more adequate, but smaller). In short, the “other” class is likely to be biased in a way that makes it more or less useless for the argument that is advanced. For instance, it includes Joyce’s Ulysses, which is of course completely unrepresentative of failing books. It has also probably been a major bestseller in the longer run.

The methods applied by the authors to compute vector representations have not been developed for the representation of the content of novels, but rather for classification of shorter documents. Again, it seems to me that the authors have made convenience decisions, which come without deeper theoretical motivation. In particular, the representations are static, whereas it seems evident that there is a time-related dynamics to narratives, which is important for the enjoyment of fiction and for bestsellerhood.

Reviewer #2: This paper aimed to study whether it is feasible to characterize and identify stories and narratives listed as best sellers by combining full-text content information and machine learning models. In this regard, the textual content of a set of books was modeled, and a series of experiments assessed the possibility of automatically differentiating a best seller from an ordinary book. In particular, the authors employed a dataset encompassing the full-text content of literary works collected from the Project Gutenberg platform.

Overall, this paper is interesting for the community of text mining and machine learning applications.

The weaknesses of this paper are as follows:

1. This paper utilized full-text content to identify best-selling books. However, there are two issues that the authors should address:

(1) The paper only employed shallow features of the full-text content, neglecting deeper features such as discourse or writing styles of the books.

(2) The authors only provided the identification results of best-selling books based on full-text content. They should also present results based on non-full-text content for comparison.

2. The dataset used in this paper consists of 219 books published between 1895 and 1924. There are two issues that the authors should address:

(1) The size of the dataset is relatively small.

(2) Many contemporary books could be utilized in this study. It is worth noting that the full-text content of current books can be accessed online. The authors should discuss this context, which differs from the past.

3. Concerning the related works in this paper, there are the following two issues:

(1) Several relevant studies have been overlooked, such as Harvey (1953), Lee et al. (2021), Lee et al. (2023), and Maity et al. (2017), among others.

Harvey, J. (1953). The content characteristics of best-selling novels. Public Opinion Quarterly, 17(1), 91-114.

Lee, S., Ji, H., Kim, J., & Park, E. (2021). What books will be your bestseller? A machine learning approach with Amazon Kindle. The Electronic Library, 39(1), 137-151.

Lee, S., Kim, J., & Park, E. (2023). Can book covers help predict bestsellers using machine learning approaches?. Telematics and Informatics, 101948.

Maity, S. K., Panigrahi, A., & Mukherjee, A. (2017, July). Book reading behavior on goodreads can predict the amazon best sellers. In Proceedings of the 2017 IEEE/ACM International Conference on Advances in Social Networks Analysis and Mining 2017 (pp. 451-454).

(2)Additionally, some related studies have employed full-text content from book tables of contents to evaluate book quality, such as Zhang & Zhou (2020).

Zhang C., Zhou Q. Assessing Books’ Depth and Breadth via Multi-level Mining on Tables of Contents. Journal of Informetrics, 2020, 14(2): 101032.

4. The methods used in this paper are relatively simple. I recommend that the authors briefly describe less important methods, while conversely, some methods should be explained in more detail, such as the classification method described in Section 5.4. Additionally, the title in Section 5.4 should be made more specific.

5. Some expressions need to be more rigorous, such as providing the full names for abbreviations that appear for the first time, like LOO.

6. The paper's structure could benefit from further refinement. It is recommended to create a separate section for related discussions, encompassing the theoretical and practical implications of this study, as well as the paper's limitations.

In summary, this study has some significance, but there are issues with the research methods and the analysis of experimental results.

6. PLOS authors have the option to publish the peer review history of their article (what does this mean?). If published, this will include your full peer review and any attached files.

Reviewer #1: No

Reviewer #2: No

---

## [Author Response · Author response to Decision Letter 0]

7 Aug 2023

Response to reviewers is also attached as a formatted PDF file.

---------

Dear Editor,

Please find attached a revised version of the manuscript ``Using Full-Text Content to Characterize and Identify Best Seller Books'', in which we have considered all the reviewers' comments. The most substantial changes made are marked in magenta in the revised version. Also attached is a list of responses to the reviewers' comments. 

Yours sincerely,

The authors.

Reviewers' comments:

Reviewer #1: 

I think this article is basically an interesting study. The “mathematical side” of the argument is sound, but crucial aspects of the study are flawed. The dataset is not adequate for the conclusions that are drawn. The main approach should be better motivated. Its limitations should be discussed in deeper theoretical detail, and the conclusions should be stated with more caution.

>>> Answer: Thank you for your comment. We agree that some of the conclusions were carelessly stated, and we have rephrased them. All the major changes made are marked in magenta in the revised version and some of these are also discussed hereon this letter.

The authors state: ``This study aims to probe whether the full-text content of the book alone can indicate if it will become a best seller.'' That would require surveying all possible methods, which isn’t feasible.

>>> Answer: Thank you for appropriately pointing this out, we have rewritten this statement as follows:

>>> ``This study aims to test whether the full-text content of the book alone can indicate if it will become a best seller.''

Same point again: ``The obtained results suggest that it is infeasible to predict the success of a literary work with high accuracy by using only its full-text content.'' But that conclusion cannot be drawn. A study of this kind can only show that the approaches that have been tested fail.

>>> Answer: Thank you for your comment, we have also changed the writing to emphasize this as a limitation in our conclusions.

>>> ``Ultimately, the results obtained from the considered approaches using only a book's full-text content were insufficient to predict the success of a literary work with high accuracy.''

The authors rightly say that ``the main trends, interests, and expectations predominating in a given period'' are crucial for bestsellerhood. This theoretical argument is a stronger one than the empirical one proposed in the paper. The empirical results are however compatible with that theoretical statement. Important factors are text-external.

>>> Answer: Thank you for your comment. We were glad to perceive the compatibility between theory and the results yielded in our research.

The conclusion “our experiments evince that the subject of the books does not seem to be a core factor for a title becoming a best seller” seems to be carelessly stated. It is hard to deny ``that the subject of the books IS A CRUCIAL FACTOR for a title becoming a best seller''. The experiments cannot disprove that. I actually find the accuracy of 0.75 surprisingly high. It would rather support the conclusion that content is very important, had the data behind it been more convincingly curated.

>>> Answer: Thank you for pointing this out. We believe that our writing and the usage of the term ``subject'' might have caused confusion between a book's genre and its content. We changed the text for more clarity, as follows:

>>> ``Nonetheless, our experiments evince that the subject (literary genre provided by Gutenberg) of a book, alone, does not seem to be enough to determine if a title will become a best seller, but rather point to the importance of content, since there are words there are more typically found in this category of books.''

The data selection is of the convenience kind. It is unclear how it would be representative of some relevant populations. The Gutenberg repository is hardly a representative selection of books. It is likely to have books interesting by their quality and/or popularity. Books belonging to typical bestseller genres which have failed to become bestsellers, say an unsuccessful crime novel, tend to be forgotten, and they are less likely to find their way to Gutenberg. If, as the authors correctly state, “the main trends, interests, and expectations” are important, bestsellers should be compared to ``fail-sellers'' in the same time and place aligning with the same trends, interests, and expectations. The size of the dataset should also be motivated.

So, “the other [class set] would have the same number of titles published in the same year — the titles randomly selected from the Gutenberg repository”. There are other relevant parameters to consider, mainly genre. But the “other” books include both literary works and non-fiction, whereas, I guess, the bestsellers to a high extent comprise genre fiction. (“the PS dataset” is more adequate, but smaller). In short, the “other” class is likely to be biased in a way that makes it more or less useless for the argument that is advanced. For instance, it includes Joyce’s Ulysses, which is of course completely unrepresentative of failing books. It has also probably been a major bestseller in the longer run.

>>> We chose to work with the Project Gutenberg repository not simply for its accessibility but due to its extensive collection of public domain books. This repository offers a diverse range of both best sellers and other preserved titles, spanning various genres and time periods. The nature of this selection underpins our analysis and allows us to conduct a broad yet meaningful comparison between best sellers and other titles that managed to be preserved over time.

>>> Concerning the size of the dataset, it is important to clarify that our constraints are dictated by the number of best sellers, which inherently are not numerous. While a larger dataset might have provided additional insights, we had to account for author-specific effects to avoid conflating book success with author popularity. This necessary control reduced the size of our dataset to its current amount.

>>> Regarding James Joyce's `Ulysses', we concur with your comment. It indeed faced numerous issues at the time of publication, which hindered its initial success. Nonetheless, our methodology has identified it as a success, echoing its subsequent recognition and popularity. While `Ulysses' may not fit the traditional best seller mold at its time of publication, its later success underscores the potential validity and foresight of our method. We added an additional discussion about this specific book in Section II of the Supplementary Material.

>>> About ``fail sellers'', if a book was completely forgotten (or never read) after its publication, it would be very difficult to find its complete content available on the internet. In fact, not even some of the best sellers listed in the consulted list were found for download. It was with this in mind that we never used the nomenclature ``fail seller'', ``failure'', or ``unsuccessful'': what we are dealing with here are the less successful (or non-successful) books in the period studied (from 1895 to 1923).

The methods applied by the authors to compute vector representations have not been developed for the representation of the content of novels, but rather for classification of shorter documents. Again, it seems to me that the authors have made convenience decisions, which come without deeper theoretical motivation. In particular, the representations are static, whereas it seems evident that there is a time-related dynamics to narratives, which is important for the enjoyment of fiction and for bestsellerhood.

>>> Answer: Thank you for your comment. We have included further explanations of our method choices for handling full-text content. As follows:

>>> ``More sophisticated techniques such as BERT and sentence BERT generate embeddings that capture richer context and semantic information of words or sentences. However, these techniques, similar to W2V and GloVe, are limited to a small number of tokens and can not be applied to large portions of texts, such as entire books. For this reason, we opted to use the doc2vec (D2V) method to extract a vector representation of each book since it has been successfully used in classification texts using large external corpora.''

Reviewer #2: 

This paper aimed to study whether it is feasible to characterize and identify stories and narratives listed as best sellers by combining full-text content information and machine learning models. In this regard, the textual content of a set of books was modeled, and a series of experiments assessed the possibility of automatically differentiating a best seller from an ordinary book. In particular, the authors employed a dataset encompassing the full-text content of literary works collected from the Project Gutenberg platform.

Overall, this paper is interesting for the community of text mining and machine learning applications.

The weaknesses of this paper are as follows:

1. This paper utilized full-text content to identify best-selling books. However, there are two issues that the authors should address:

(1) The paper only employed shallow features of the full-text content, neglecting deeper features such as discourse or writing styles of the books.

(2) The authors only provided the identification results of best-selling books based on full-text content. They should also present results based on non-full-text content for comparison.

>>> Answer: Thank you for your comment. We have performed a new experiment where we address non-full-text content. Its discussion is in Section I of the Supplementary Material provided. Additionally, we have also included in Section III of the Supplementary Material a new discussion concerning readability scores and other textual features.

2. The dataset used in this paper consists of 219 books published between 1895 and 1924. There are two issues that the authors should address:

(1) The size of the dataset is relatively small.

(2) Many contemporary books could be utilized in this study. It is worth noting that the full-text content of current books can be accessed online. The authors should discuss this context, which differs from the past.

>>> Answer: Thank you for your comment. Unfortunately, as we discussed in the text (Section 4, copied below), the size and the publication time of the books in the dataset is, indeed, one of the limitations of this work. We recognize how this can restrict our research, but we took careful measures to work with the data that was publicly available and we believe that our work yields valuable and valid results.

>>> ``Additionally, it is worth mentioning that some factors were imperative in the limited number of books of the dataset (namely, 219 instances). First, we adhere to titles in the public domain only. Although there are discussions about the fair use of such content in scientific works, there is no consensus on the validity of using copyrighted pieces. Second, we considered only one book from each author to avoid identification of authorship by machine learning algorithms to be applied later. Third, because one of the design decisions was to work with a balanced database, the number of bestsellers becomes a limiting factor for the number of non-bestselling books. Lastly, we collected the same number of successes and non-successes per year of publication (which even led to one less non-successful book due to the unavailability of another title in one of the years considered). We emphasize, nonetheless, that such a temporal factor is essential because there will always be a possibility that titles from different periods may be very distinct in terms of content and writing style.''

>>> We highlight the excerpt ``Although there are discussions about the fair use of such content in scientific works, there is no consensus on the validity of using copyrighted pieces''. While it would be possible, yes, to obtain the complete contents of more contemporary books, this would be ethically (and legally) debatable.

3. Concerning the related works in this paper, there are the following two issues:

(1) Several relevant studies have been overlooked, such as Harvey (1953), Lee et al. (2021), Lee et al. (2023), and Maity et al. (2017), among others.

Harvey, J. (1953). The content characteristics of best-selling novels. Public Opinion Quarterly, 17(1), 91-114.

Lee, S., Ji, H., Kim, J., & Park, E. (2021). What books will be your bestseller? A machine learning approach with Amazon Kindle. The Electronic Library, 39(1), 137-151.

Lee, S., Kim, J., & Park, E. (2023). Can book covers help predict bestsellers using machine learning approaches?. Telematics and Informatics, 101948.

Maity, S. K., Panigrahi, A., & Mukherjee, A. (2017, July). Book reading behavior on goodreads can predict the amazon best sellers. In Proceedings of the 2017 IEEE/ACM International Conference on Advances in Social Networks Analysis and Mining 2017 (pp. 451-454).

(2)Additionally, some related studies have employed full-text content from book tables of contents to evaluate book quality, such as Zhang & Zhou (2020).

Zhang C., Zhou Q. Assessing Books’ Depth and Breadth via Multi-level Mining on Tables of Contents. Journal of Informetrics, 2020, 14(2): 101032.

>>> Answer: Thank you for your comment, we have included the references in our discussion.

4. The methods used in this paper are relatively simple. I recommend that the authors briefly describe less important methods, while conversely, some methods should be explained in more detail, such as the classification method described in Section 5.4. Additionally, the title in Section 5.4 should be made more specific.

>>> Answer: Thank you for pointing this out. We have changed this section's title and added more discussion on both the classification methods used and the classification problem we are addressing, as follows:

>>> ``The identification and classification of textual patterns were performed using traditional well-known machine learning classifiers. We considered different classifier strategies, including $k$-nearest neighbors (KNN) (based on the probable similarity of nearest neighbors), naive Bayes (NB) (that estimates the class-conditional probability based on the Bayes theorem and assuming conditional independence between attributes), decision tree (DT) (which classifies an example of the test record based on a series of discriminating questions about its attributes), support-vector machine (SVM) (based on finding hyper-planes that can linearly separate data - called support vectors), and, finally, the two that yielded the best results: random forest (RF) and logistic regression (LR).

>>> In just a few words, Random Forest is a class of ensemble methods designed over DT classifiers. It uses multiple decision trees, built using a set of random vectors, combining each of their predictions to yield a final classification.

On the other hand, Logistic Regression is based on determining the conditional probability of an event happening. It models this probability by minimizing a negative likelihood function for the labeled classes.''

5. Some expressions need to be more rigorous, such as providing the full names for abbreviations that appear for the first time, like LOO.

>>> Answer: Thank you for your comment. We have carefully corrected these mistakes in the text.

6. The paper's structure could benefit from further refinement. It is recommended to create a separate section for related discussions, encompassing the theoretical and practical implications of this study, as well as the paper's limitations.

>>> Answer: Thank you for your comment. We have included a separate supplementary material to complement some valuable discussions about our research. We also included a final paragraph in the Conclusion section of the paper to address the implications of our study, as follows:

>>> ``Regardless of the dataset limitations we recognized in this research, we have employed careful effort to minimize any undesirable impact of external factors, such as authorship, publication pe

---

## [Decision Letter · Decision Letter 1]

10 Oct 2023

PONE-D-23-04137R1Using Full-Text Content to Characterize and Identify Best Seller BooksPLOS ONE

Dear Dr. Silva,

Thank you for submitting your manuscript to PLOS ONE. After careful consideration, we feel that it has merit but does not fully meet PLOS ONE’s publication criteria as it currently stands. Therefore, we invite you to submit a revised version of the manuscript that addresses the points raised during the review process.

We look forward to receiving your revised manuscript.

Kind regards,

Heba El-Fiqi

Academic Editor

PLOS ONE

Reviewers' comments:

Reviewer's Responses to Questions

**Comments to the Author**

1. If the authors have adequately addressed your comments raised in a previous round of review and you feel that this manuscript is now acceptable for publication, you may indicate that here to bypass the “Comments to the Author” section, enter your conflict of interest statement in the “Confidential to Editor” section, and submit your "Accept" recommendation.

Reviewer #2: (No Response)

Reviewer #3: (No Response)

2. Is the manuscript technically sound, and do the data support the conclusions?

Reviewer #2: Yes

Reviewer #3: Partly

3. Has the statistical analysis been performed appropriately and rigorously? 

Reviewer #2: Yes

Reviewer #3: Yes

4. Have the authors made all data underlying the findings in their manuscript fully available?

Reviewer #2: Yes

Reviewer #3: Yes

5. Is the manuscript presented in an intelligible fashion and written in standard English?

Reviewer #2: Yes

Reviewer #3: Yes

6. Review Comments to the Author

Reviewer #2: The dataset used in this paper consists of 219 books published between 1895 and 1924. The title of this paper is "Using Full-Text Content to Characterize and Identify Best Seller Books". Therefore, one issue that needs to be addressed is whether the method in this paper is applicable to identification of current Best Seller Books. This is because the full-text content of current books can be accessed online. Therefore, the author needs to limit the title of the paper and conduct necessary discussions.

Reviewer #3: Pros:

1- The paper is organized and well written.

2- This paper shows a comparison analysis between the proposed model with numerous traditional classifiers.

3- The proposed method and results help publishing companies and writers.

Cons:

1- The results were not good (the average accuracy of 75%). These results may make the model’s decisions untrusted.

2- It may be better to check deep learning like BERT or a large language model (LLM).

3- The pictures are blurring.

Note:

It may be better to remove “Then” in the text “Then, to obtain quantitative and more objective results, we employed various classifiers”

7. PLOS authors have the option to publish the peer review history of their article (what does this mean?). If published, this will include your full peer review and any attached files.

Reviewer #2: No

Reviewer #3: **Yes: **Hashim Abu-Gellban

---

## [Author Response · Author response to Decision Letter 1]

21 Nov 2023

Dear Editor,

Please find attached a revised version of the manuscript “Using Full-Text Content to Characterize and Identify Best Seller Books”, in which we have considered all the reviewers’ comments. The most substantial changes made are marked in magenta in the revised version. Also attached is a list of re- sponses to the reviewers’ comments.

Yours sincerely, The authors.

Reviewers’ comments:

Reviewer #2:

The dataset used in this paper consists of 219 books published between 1895 and 1924. The title of this paper is ”Using Full-Text Content to Charac- terize and Identify Best Seller Books”. Therefore, one issue that needs to be addressed is whether the method in this paper is applicable to identification of current Best Seller Books. This is because the full-text content of current books can be accessed online. Therefore, the author needs to limit the title of the paper and conduct necessary discussions.

Answer: Thanks for pointing that out. We updated the paper title to “Using full-text content to identify best sellers: a study of early 20th-century literature”. In this way, we hope to reflect the period of the books analyzed, ensuring that the reader is aware of this information before starting to read. Furthermore, we added a paragraph in the Conclusions section that reaffirms that one of the limitations of this work is that we cannot analyze books pub- lished after the beginning of the 20th century due to copyright reasons. The paragraph is as follows:

(...) Concerning the limitations of the work, the two main points we stress are (i) the absence of modern books in the database and (ii) the absence of more modern modeling techniques. As previously discussed, the scarcity of books is due to copyright laws that protect the complete contents of modern books. Even though such content is found free of charge on the internet, we do not have the right to use it.

Reviewer #3:

Pros:

1. The paper is organized and well written.

2. This paper shows a comparison analysis between the proposed model with numerous traditional classifiers.

3. The proposed method and results help publishing companies and writ- ers.

Cons:

1. The results were not good (the average accuracy of 75%). These results may make the model’s decisions untrusted.

Answer: Thanks for pointing that out. We recognize that 75% accu- racy is not a lot, but we highlight the intricate nature of the problem, which makes it difficult to achieve high accuracies (such as 80 or 90%). To make this clear, we added an excerpt in the Conclusions section. The excerpt is as follows:

(...) In addition to these points, it is also worth mentioning that although an accuracy greater than 80 or 90% would be desirable, it would be un- realistic to predict the success of works with such a high outcome, given the intricate and multifaceted nature of such a task. Factors like mar- keting and trends can influence the popularity of books in ways difficult to predict or measure. Therefore, the 75% accuracy result becomes rea- sonable, although somewhat limited, if we think we are exploring solely the textual content of each book.

2. It may be better to check deep learning like BERT or a large language model (LLM).

Answer: Thanks for pointing that out. In Text embeddings subsection we mention the following:

(...) More sophisticated techniques such as BERT and sentence BERT generate embeddings that capture richer context and semantic informa- tion of words or sentences. However, these techniques, similar to W2V and GloVe, are limited to a small number of tokens and can not be applied to large portions of texts, such as entire books.

However, to highlight this limitation (which may not have become so clear during reading), we added a new excerpt in a Conclusions section paragraph that mentions the limitations of the work. The excerpt is as follows:

(...) Regarding modeling, more modern techniques, such as BERT, do not deal well with long texts. As the median size of our dataset is approximately 90,000 characters, it would not be appropriate to apply such a technique. Even if the modeling yielded highly accurate results, they would not be reliable.

3. The pictures are blurring.

Answer: Thanks for your comment. We double checked, and the im- ages are in good quality. The PLOS ONE submission system automatically converts the images from the PDF to lower quality versions. To access the figures in higher resolution, click on “download figure” in the PDF.

Note: It may be better to remove “Then” in the text “Then, to obtain quantitative and more objective results, we employed various classifiers”

Answer: Indeed, thanks for spotting it. We updated it accordingly.

---

## [Decision Letter · Decision Letter 2]

12 Dec 2023

PONE-D-23-04137R2Using full-text content to characterize and identify best seller books: a study of early 20th-century literaturePLOS ONE

Dear Dr. Silva,

Thank you for submitting your manuscript to PLOS ONE. After careful consideration, we feel that it has merit but does not fully meet PLOS ONE’s publication criteria as it currently stands. Therefore, we invite you to submit a revised version of the manuscript that addresses the points raised during the review process.

We look forward to receiving your revised manuscript.

Kind regards,

Heba El-Fiqi

Academic Editor

PLOS ONE

Reviewers' comments:

Reviewer's Responses to Questions

**Comments to the Author**

1. If the authors have adequately addressed your comments raised in a previous round of review and you feel that this manuscript is now acceptable for publication, you may indicate that here to bypass the “Comments to the Author” section, enter your conflict of interest statement in the “Confidential to Editor” section, and submit your "Accept" recommendation.

Reviewer #2: All comments have been addressed

Reviewer #3: (No Response)

2. Is the manuscript technically sound, and do the data support the conclusions?

Reviewer #2: Yes

Reviewer #3: Partly

3. Has the statistical analysis been performed appropriately and rigorously? 

Reviewer #2: Yes

Reviewer #3: Yes

4. Have the authors made all data underlying the findings in their manuscript fully available?

Reviewer #2: Yes

Reviewer #3: Yes

5. Is the manuscript presented in an intelligible fashion and written in standard English?

Reviewer #2: Yes

Reviewer #3: Yes

6. Review Comments to the Author

Reviewer #2: Thank you for your revision. After the revision, the motivation for the research is clearly stated, the research methods are appropriate, and the research results are credible. I believe this paper has satisfactorily answered my previous doubts. The paper can be accepted for publication.

Reviewer #3: Pros:

1- The paper is organized and well written.

2- This paper shows a comparison analysis between the proposed model with numerous traditional classifiers.

3- The proposed method and results help publishing companies and writers.

Cons:

1- The obtained results, with an average accuracy of 75%, fall short of expectations, potentially undermining the reliability of the model's decisions. Despite the author's assertion that the results were reasonable, they lie in the middle ground between randomness (50%) and high accuracy (<99%). Additionally, depending solely on the accuracy metric is insufficient. The paper should include other evaluation metrics, such as precision, recall, and F1-score, to provide a more comprehensive assessment of the classification models.

2- Exploring deep learning models like BERT or a large language model (LLM) could be more beneficial. The revised paper (R2) notes that "BERT does not deal well with long texts," referencing a study that utilized only the first 510 tokens of extensive text. However, this truncation limits classifiers' awareness of most the input text. There are alternative techniques to handle lengthy text when applying BERT, such as segmenting the text into chunks to fit the model's input size. Moreover, LLMs exhibit a capability to manage larger input sizes.

3- The dataset limitation is apparent, consisting of only 219 examples.

7. PLOS authors have the option to publish the peer review history of their article (what does this mean?). If published, this will include your full peer review and any attached files.

Reviewer #2: No

Reviewer #3: No

---

## [Author Response · Author response to Decision Letter 2]

9 Mar 2024

Reviewers’ comments:

Reviewer #3:

Pros:

1. The paper is organized and well written.

2. This paper shows a comparison analysis between the proposed model with numerous traditional classifiers.

3. The proposed method and results help publishing companies and writers.

Cons:

1. The obtained results, with an average accuracy of 75%, fall short of expectations, potentially undermining the reliability of the model’s decisions. Despite the author’s assertion that the results were reasonable, they lie in the middle ground between randomness (50%) and high accuracy (<99%). Additionally, depending solely on the accuracy metric is insufficient. The paper should include other evaluation metrics, such as precision, recall, and F1-score, to provide a more comprehensive assessment of the classification models.

Answer: Thank you for your comment. The proposed model aims to tackle the very complex task of predicting the success of a book solely based on its content. While a 75% accuracy rate may seem lower in terms of reliability, for instance, for deployment in the industry, we never anticipated the model would achieve high values anyway. We acknowledge that several external factors, beyond just the content, significantly influence whether a book becomes a bestseller. This includes the author popularity, adaptations into other media forms (e.g., movies inspired by the book), and the political and social context at the time of publication. Such an accuracy value showcases that the content seems to be an important factor in determining the success of books. Future work may improve these numbers by employing more sophisticated methods. For delivering the prediction task, future models could also incorporate content, author information, and more context in order to achieve better performance. We included such a suggestion in the conclusions.

Concerning the other evaluation metrics, we added a new section (Section IV - Assessing precision, recall, and f1-score metrics) in the supplementary material, where we expose the precision, recall, and f1-score for the modeling/configuration that achieved the best-obtained results. As stated there, these metrics are not considerably gainful, as they yield results too similar to the accuracy. However, we recognize the importance of exposing them so that the reader can understand that (i) accuracy is an appropriate metric for this case and (ii) all the other metrics sustain the value of the accuracy.

2. Exploring deep learning models like BERT or a large language model (LLM) could be more beneficial. The revised paper (R2) notes that ”BERT does not deal well with long texts,” referencing a study that utilized only the first 510 tokens of extensive text. However, this truncation limits classifiers’ awareness of most the input text. There are alternative techniques to handle lengthy text when applying BERT, such as segmenting the text into chunks to fit the model’s input size. Moreover, LLMs exhibit a capability to manage larger input sizes.

Answer: Indeed, there are alternative methodologies in the literature, such as aggregating vector representations or using larger token sizes. However, even these approaches are constrained by a relatively small token limit, extending up to 32,000 tokens (e.g., the Longformer [1] and TransformerXL [2]). Considering that a typical book may contain between 70,000 and 120,000 tokens, these models still fall short of covering entire texts. Moreover, the inherent computational memory requirements for processing with large models add another layer of complexity. Additionally, pre-trained large language models may not be ideally suited for our analysis due to potential biases in their training datasets. For instance, a preliminary examination of ChatGPT revealed its ability to detect whether a book was a bestseller based solely on its title, indicating possible data contamination. Given our lack of control over the training datasets of these large language models, their applicability in our study is not clear.

[1] Beltagy, Iz, Matthew E. Peters, and Arman Cohan. ”Longformer:

The long-document transformer.” arXiv preprint arXiv:2004.05150 (2020). [2] Dai, Z., Yang, Z., Yang, Y., Carbonell, J., Le, Q. V., & Salakhutdinov, R. (2019). Transformer-xl: Attentive language models beyond a fixed-length context. arXiv preprint arXiv:1901.02860

Although fine-tuning and exploring the performance of large language models for predicting the success of books is beyond the current scope of our research, we propose in the conclusions that future studies should assess the efficacy of LLMs for this specific task. Two significant challenges must be addressed: the limited token size, which might be mitigated through a strategy of piecewise summarization, and ensuring the training dataset used in the model does not influence its analysis. One potential solution could involve using only inputs from books published after the model training, which poses challenges due to restricted access to contemporary works. Alternatively, models could be carefully trained with datasets excluding references to the analyzed books.

3. The dataset limitation is apparent, consisting of only 219 examples.

Answer: Thanks for your comment, we really appreciate that. We acknowledge the limitation of our dataset and added a new excerpt on the Conclusions to reinforce this restriction. The excerpt is as follows:

(...) Concerning the limitations of the work, the three main points we stress are (i) the absence of modern books in the database, (ii) the absence of more modern modeling techniques, and (iii) the limitation in dataset size imposed by the number of available best-selling books. (...) We are restricted by the number of books listed as best sellers and also available in the public domain. As previously stated, we can not leverage books that don’t have free content. Also, best-selling books are scarce per nature: if all books were best-selling pieces, this study would not even exist.

---

## [Editor Report · Decision Letter 3]

27 Mar 2024

Using full-text content to characterize and identify best seller books: a study of early 20th-century literature

PONE-D-23-04137R3

Dear Dr. Silva,

We’re pleased to inform you that your manuscript has been judged scientifically suitable for publication and will be formally accepted for publication once it meets all outstanding technical requirements.

Kind regards,

Heba El-Fiqi

Academic Editor

PLOS ONE